# Differential Diagnosis of Hyperferritinemia in Critically Ill Patients

**DOI:** 10.3390/jcm12010192

**Published:** 2022-12-27

**Authors:** Friederike S. Schuster, Peter Nyvlt, Patrick Heeren, Claudia Spies, Moritz F. Adam, Thomas Schenk, Frank M. Brunkhorst, Gritta Janka, Paul La Rosée, Cornelia Lachmann, Gunnar Lachmann

**Affiliations:** 1Department of Anesthesiology and Operative Intensive Care Medicine (CCM, CVK), Charité–Universitätsmedizin Berlin, Freie Universität Berlin and Humboldt-Universität zu Berlin, 13353 Berlin, Germany; 2Department of Hematology and Oncology, Universitätsklinikum Jena, 07743 Jena, Germany; 3Department of Anesthesiology and Intensive Care Medicine, Center for Clinical Studies, Universitätsklinikum Jena, 07743 Jena, Germany; 4Clinic of Pediatric Hematology and Oncology, University Medical Center Eppendorf, 20246 Hamburg, Germany; 5Klinik für Innere Medizin II, Schwarzwald-Baar-Klinikum, 78052 Villingen-Schwenningen, Germany; 6Berlin Institute of Health (BIH) at Charité–Universitätsmedizin Berlin, 10178 Berlin, Germany

**Keywords:** hemophagocytic lymphohistiocytosis (HLH), macrophage activation syndrome (MAS), hemophagocytic syndrome (HS), critically ill patients, ferritin, sepsis, liver disease, hematological malignancy, differential diagnosis

## Abstract

Background: Elevated serum ferritin is a common condition in critically ill patients. It is well known that hyperferritinemia constitutes a good biomarker for hemophagocytic lymphohistiocytosis (HLH) in critically ill patients. However, further differential diagnoses of hyperferritinemia in adult critically ill patients remain poorly investigated. We sought to systematically investigate hyperferritinemia in adult critically ill patients without HLH. Methods: In this secondary analysis of a retrospective observational study, patients ≥18 years admitted to at least one adult intensive care unit at Charité–Universitätsmedizin Berlin between January 2006 and August 2018, and with hyperferritinemia of ≥500 μg/L were included. Patients with HLH were excluded. All patients were categorized into non-sepsis, sepsis, and septic shock. They were also classified into 17 disease groups, based on their ICD-10 codes, and pre-existing immunosuppression was determined. Uni- and multivariable linear regression analyses were performed in all patients. Results: A total of 2583 patients were analyzed. Multivariable linear regression analysis revealed positive associations of maximum SOFA score, sepsis or septic shock, liver disease (except hepatitis), and hematological malignancy with maximum ferritin. T/NK cell lymphoma, acute myeloblastic leukemia, Kaposi’s sarcoma, acute or subacute liver failure, and hepatic veno-occlusive disease were positively associated with maximum ferritin in post-hoc multivariable linear regression analysis. Conclusions: Sepsis or septic shock, liver disease (except hepatitis) and hematological malignancy are important differential diagnoses in hyperferritinemic adult critically ill patients without HLH. Together with HLH, they complete the quartet of important differential diagnoses of hyperferritinemia in adult critically ill patients. As these conditions are also related to HLH, it is important to apply HLH-2004 criteria for exclusion of HLH in hyperferritinemic patients. Hyperferritinemic critically ill patients without HLH require quick investigation of differential diagnoses.

## 1. Introduction

Ferritin is released from various cell types depending on iron levels, as part of the oxidative stress response [1,2], and by macrophages in pro-inflammatory conditions [3]. Ferritin is, therefore, not only a marker of iron status, but also of inflammation [4]. In several diseases and underlying conditions, ferritin may be altered: iron deficiency anemia [5], hemochromatosis [6], and rheumatological diseases [7,8,9,10]. In general ward patients, hyperferritinemia was associated with metabolic syndrome [11], alcohol consumption [12], kidney disease, liver disease, infection, hematological malignancy, hemolytic anemia, and hemophagocytic lymphohistiocytosis (HLH) [13,14]. Hyperferritinemia was found to be predictive for in-hospital mortality in various contexts [15,16,17,18]. It has also gained attention during the SARS-CoV-2 pandemic [19] and in the context of hyperferritinemic sepsis. The latter is part of hyperferritinemic syndromes, which also contain HLH, adult-onset Still’s disease, catastrophic antiphospholipid syndrome, and multi-inflammatory syndrome related to COVID-19 [20,21].

It is well known that hyperferritinemia is a good biomarker for HLH in adult critically ill patients. In a previous study of 2623 adult critically ill patients, we found a ferritin cutoff of 9083 µg/L to be sensitive and specific for the diagnosis of HLH (92.5% sensitivity, 91.9% specificity) [22]. Out of 954 non-sepsis and non-HLH patients, those with varicella-zoster virus (VZV), hepatitis, and malaria showed the highest ferritin levels. Further studies of hyperferritinemia in critically ill patients found high ferritin values in patients with infection, iron overload [13,14,23,24], hematological and solid malignancy [13,14,23], rheumatologic/inflammatory disease [13,14,24], renal failure [13,14], hemolytic anemia or acute hemolysis [14,24], liver dysfunction [13,23,24], and cytokine release syndrome [24]. However, these studies were merely descriptive without any multivariable analysis, leading to a high risk of bias due to various diseases overlapping in patients. Currently, there is only one study available, which investigated hyperferritinemia in adult critically ill patients using multivariable analysis. Schram et al. [14] found hemolytic anemia to be associated with hyperferritinemia. However, they studied only 113 patients with extreme hyperferritinemia of >50,000 µg/L, who were not further categorized into intensive care unit (ICU) or non-ICU patients. Hence, the available descriptive research and the consecutive lack of systematic analysis of differential diagnoses in hyperferritinemic critically ill patients leaves a high grade of uncertainty for intensive care physicians when interpreting high ferritin levels. Given the high diagnostic value of ferritin for HLH diagnosis, as well as the lack of systematic analysis of hyperferritinemic critically ill patients without HLH, we sought to identify important differential diagnoses of hyperferritinemia in adult critically ill patients without HLH. Clarification of differential diagnoses of hyperferritinemia will improve diagnostic workup in critical care.

## 2. Methods

The study was registered with www.ClinicalTrials.gov (NCT02854943) on 1 August 2016.

### 2.1. Patients

This secondary analysis of a retrospective observational study [22] was carried out at Charité–Universitätsmedizin Berlin. All critically ill patients admitted to at least one adult surgical, anesthesiological or medical ICU between January 2006 and August 2018 were reviewed. Data extraction was performed using two electronic data management systems at Charité–Universitätsmedizin Berlin (COPRA, Sasbachwalden, Germany and SAP, Walldorf, Germany). The study period was defined from ICU admission until discharge, transfer or death. Inclusion criteria were age ≥18 years, at least one ferritin value available during ICU stay, and hyperferritinemia of at least 500 µg/L, according to HLH-2004 criteria [25]. For the purpose of this secondary analysis, patients with HLH [17] were excluded. The diagnosis of HLH containing expert reviews was previously described in detail [17,22,26]. In patients with multiple ferritin measurements, the maximum ferritin was considered for further analyses. For descriptive analyses, patients were divided into the hyperferritinemia (maximum ferritin <9083 µg/L) and extreme hyperferritinemia (maximum ferritin ≥9083 µg/L) groups, based on the optimal ferritin cutoff found for diagnosis of HLH in adult critically ill patients [22].

### 2.2. Diagnosis of Sepsis, Septic Shock and Other Diagnoses

Non-sepsis, sepsis, and septic shock patients were defined by ICD-10 codes. All patients were further classified into 17 disease groups, based on their ICD-10 codes: liver disease (except hepatitis), renal disease, autoimmune disease, hepatitis, tuberculosis, human immunodeficiency virus (HIV), herpes simplex virus (HSV), cytomegalovirus (CMV), Epstein-Barr virus (EBV), VZV, influenza, malaria, (bacterial/viral/fungal) infection, inflammation without infection, hematological malignancy, solid tumors, and history of (stem cell/organ) transplantation. The ICD-10 codes of sepsis, septic shock and all 17 disease groups with corresponding numbers of patients are shown in Appendix A. Pre-existing immunosuppression was defined as confirmed diagnosis of HIV or long-term treatment with conventional immunosuppressants or antibodies (e.g., Adalimumab, Anakinra, Everolimus), prior or ongoing (radio-)chemotherapy, prior splenectomy or stem cell/organ transplantation, or primary immunodeficiency.

### 2.3. Data Collection of Diagnostic Markers

Of all patients, data was extracted for blood counts, triglycerides, fibrinogen, international normalized ratio (INR), activated partial thromboplastin time (aPTT), aspartate aminotransferase (AST), alanine aminotransferase (ALT), bilirubin, gamma glutamyl transferase (ɣGT), alkaline phosphatase (AP), albumin, creatinine, c reactive protein (CRP), procalcitonin (PCT), lactate, lactate dehydrogenase (LDH), and maximum core body temperature. All parameters were taken at the day of maximum ferritin measurement or during a plausible time range, if no value was documented that day (Appendix A).

### 2.4. Statistical Analysis

Results are shown as median ± quartiles, or percentage, respectively. Differences between the patient groups were calculated using Mann-Whitney U test for continuous data and the chi-square test for qualitative data. In the first step, we performed univariable linear regression analyses separately for each dichotomous (no/yes) variable of interest (17 disease groups and pre-existing immunosuppression), to analyze the influence on maximum ferritin. All significant variables were entered into a multivariable linear regression model, adjusting for age, sex, body mass index (BMI), sepsis or septic shock (into categorical variable non-sepsis/sepsis without shock/septic shock), and maximum sequential organ failure assessment (SOFA) score to analyze the influence on maximum ferritin. As a post-hoc analysis, we first conducted separate univariable linear regression analyses for each ICD-10 code (dichotomous (no/yes)), which corresponded to the positively associated disease groups in the primary multivariable linear regression model, to analyze the influence on maximum ferritin. Then, we reran the primary multivariable linear regression model again, by replacing the positively associated disease groups with their corresponding ICD-10 codes (each as a dichotomous (no/yes) variable), which were significantly associated with maximum ferritin in the former univariable regression analyses. All regression analyses were performed using the full cohort of patients. SPSS^®^ Statistics, version 26.0 software (IBM Corporation, Armonk, NY, USA) was used for all statistical analyses, except for depiction of the Figures, for which we used The R Project for Statistical Computing (version 4.2.2). A *p* value of < 0.05 was considered statistically significant.

## 3. Results

### 3.1. Study Population and Characteristics

Of 116,310 patients admitted to the ICUs, 6340 patients had at least one ferritin value available during their ICU stay, of whom 2623 had hyperferritinemia of ≥500 µg/L [22,26]. After exclusion of 40 patients with HLH [17], 2583 patients were finally analyzed: 2373 patients with hyperferritinemia <9083 µg/L and 210 patients with extreme hyperferritinemia ≥9083 µg/L (Figure 1). The distribution of ferritin measurements over time is depicted in Appendix A.

Patients with extreme hyperferritinemia were younger, more often female, had more septic shock, higher rates of hepatitis, liver disease, history of stem cell/organ transplantation, CMV infection, hematological malignancy, and pre-existing immunosuppression (Table 1). These patients showed lower rates of inflammation without infection, lower platelet count and fibrinogen, higher triglycerides, INR, aPTT, AST, ALT, bilirubin, yGT, AP, CRP, PCT, lactate, LDH, and maximum core body temperature, a higher rate of hemodialysis, higher SOFA scores, a lower ICU and inpatient duration, and a higher mortality rate (Table 1).

### 3.2. Underlying Diseases Associated with Hyperferritinemia

Univariable linear regression analyses found pre-existing immunosuppression, liver disease (except hepatitis), history of stem cell/organ transplantation, and hematological malignancy to be associated with maximum ferritin (Appendix A). Multivariable linear regression analysis revealed positive associations of maximum SOFA score, sepsis or septic shock, liver disease, and hematological malignancy with maximum ferritin, of which liver disease had the highest effect, followed by hematological malignancy (Table 2). Maximum ferritin levels between the disease groups are shown in Figure 2, and in Figure 3 between non-sepsis, sepsis, and septic shock patients, to also give an overview for patients without liver disease and hematological malignancy.

In the post-hoc analysis, univariable linear regression analyses found registration for high-urgency liver transplantation, acute, subacute and chronic liver failure, hepatic encephalopathy (grade 1–4, and unspecified), hepatic veno-occlusive disease, history of stem cell/organ transplantation, T/NK cell lymphoma, acute lymphocytic and acute myeloblastic leukemia, Kaposi’s sarcoma, and aplastic anemia to be associated with maximum ferritin (Appendix A). Post-hoc multivariable linear regression analysis revealed positive associations of acute or subacute liver failure, hepatic veno-occlusive disease, T/NK cell lymphoma, acute myeloblastic leukemia, and Kaposi’s sarcoma with maximum ferritin, of which T/NK cell lymphoma had the highest effect, followed by hepatic veno-occlusive disease, and Kaposi’s sarcoma (Table 3). Maximum ferritin levels between the positively associated subgroups of liver disease and hematological malignancy are shown in Figure 4.

### 3.3. Description of Disease Severity and Hyperferritinemia

For description of disease severity, we considered AST as a diagnostic marker for liver disease and LDH as a diagnostic marker for hematological malignancy. Patients with higher AST levels in liver disease and higher LDH levels in hematological malignancy showed higher maximum ferritin (Figure 5).

## 4. Discussion

This is the first study that systematically investigated differential diagnoses of hyperferritinemia in adult critically ill patients without HLH using multivariable analysis. We observed positive associations of sepsis or septic shock, liver disease (except hepatitis), and hematological malignancy with hyperferritinemia. In a post-hoc multivariable analysis, T/NK cell lymphoma, acute myeloblastic leukemia, Kaposi’s sarcoma, acute or subacute liver failure, and hepatic veno-occlusive disease showed positive associations with maximum ferritin. When sepsis or septic shock, liver disease, and hematological malignancy were present in combination, ferritin levels were further increased in the descriptive analysis. Increasing disease severity also showed higher ferritin levels in the descriptive analysis.

As differential diagnoses of hyperferritinemia remained poorly investigated in critically ill patients, there is a high grade of uncertainty in interpreting high ferritin levels. For instance, HLH was found undiagnosed in the majority of adult critically ill patients [27]. Our results show that sepsis or septic shock, liver diseases, and hematological malignancies are the main factors associated with hyperferritinemia in adult critically ill patients without HLH. The highest effect was seen for liver disease, followed by hematological malignancy and sepsis or septic shock, which implicates liver disease as the main driver of hyperferritinemia in adult critically ill patients without HLH. Besides HLH, they need consideration as differential diagnoses in hyperferritinemic critically ill patients. Therefore, our results imply two major points for clinical practice: firstly, HLH-2004 criteria need to be applied in hyperferritemic patients to exclude HLH, especially in the context of liver disease, hematological malignancy and sepsis or septic shock, as these conditions are all related to HLH. In particular, hematological malignancy may not only be the underlying condition for hyperferritinemia, but can also be the underlying trigger in HLH patients [28]. Similarly, liver disease may either be caused by HLH [28] or increase ferritin levels itself. HLH-2004 criteria can safely differentiate between HLH and non-HLH as causes of hyperferritinemia in adult critically ill patients [26]. Secondly, hyperferritinemic critically ill patients without HLH need further diagnostic workup for differential diagnoses, i.e., liver disease, hematological malignancy, and sepsis or septic shock. Life-threatening diseases of the hyperferritinemic syndromes also need consideration. Our study helps intensive care physicians to systematically approach the diagnostic challenge of hyperferritinemia. As hyperferritinemic critically ill patients are associated with high mortality rates [22], quickly finding the underlying condition of hyperferritinemia is of the utmost importance.

Among the disease group of hematological malignancy, T/NK cell lymphoma had the highest effect on maximum ferritin. Ferritin values are assumed to correspond to the level of inflammation in cancer [29], and to the proliferation rates of T/NK cell lymphomas [30]. Other reasons for increasing ferritin levels in hematological malignancies might be ferritin production of malignant cells themselves [31], or decreased hematopoiesis with consecutive irregular storage of unused iron within the bone marrow [32,33]. However, the exact underlying pathomechanism of hyperferritinemia in hematological malignancies remains unclear and further investigation is needed. Elevation of LDH is frequently seen in hematological malignancies [34,35], which is why we assumed LDH as a disease severity marker and observed higher LDH levels accompanied by higher ferritin values during descriptive analysis. Thus, ferritin might also constitute an indicator of disease severity in hematological malignancy.

Hyperferritinemia in liver disease may result from damaged liver cells [4], especially from hepatocytes, which contain high amounts of iron, and also synthesize ferritin [36]. Liver disease patients with higher AST had higher ferritin levels, at least in descriptive statistics, leading to the assumption that disease severity is an additional influencing factor on hyperferritinemia, and ferritin might be a marker of disease severity. Previous studies have also shown associations between liver diseases and hyperferritinemia, each in the context of chronic liver dysfunction [13], non-alcoholic steatohepatitis [37,38,39], hemochromatosis, [38] alcohol-related liver disease, and hepatitis C [39]. However, our study analyzed hepatitis as its own entity, not as part of liver disease, and found no association with hyperferritinemia, which might be explained by inflammatory, rather than cell-damaging, processes.

Sepsis or septic shock, as well as the corresponding maximum SOFA score, is the third major factor that was associated with maximum ferritin levels. These conditions come along with metabolic disorders, in which cells produce higher levels of reactive oxygen species [40], leading to the expression of ferritin as an antioxidative stress response [41] and as an acute phase reactant [42]. Hyperferritinemia is also related to a pro-inflammatory state in sepsis, with increases of interleukin (IL)-6, IL-18, Interferon γ, sCD163 and a decrease of the IL-10/tumor necrosis factor α ratio as quantitative markers of inflammation [43]. In the present analysis, we additionally found maximum ferritin levels in liver disease and hematological malignancy to be higher, if co-occurring with sepsis or septic shock. This highlights that several conditions can increase ferritin simultaneously, making the diagnosis in hyperferritinemic patients more complex. Lately, hyperferritinemic sepsis has been an emerging object of research [44]. The terms ‘hyperinflammatory sepsis’, ‘macrophage activation syndrome (MAS)-like sepsis’, and ‘hyperferritinemic sepsis’ refer to sepsis patients who do not completely fulfil HLH-2004 criteria but have high mortality rates [43]. This particular cohort needs special consideration in future studies to develop targeted sepsis therapies.

It is important to note that ferritin in adult critically ill patients is only predictive for HLH; not for sepsis or septic shock, liver disease nor hematological malignancy [22]. Maximum ferritin of sepsis or septic shock, liver disease and hematological malignancy was much lower compared to median levels of maximum ferritin of adult critically ill HLH patients (31,674 µg/L) as shown in [22]. In this regard, ferritin serves as a sensitive and specific screening marker of HLH in critically ill patients, and as an indicator of severe diseases, such as sepsis or septic shock, liver disease, and hematological malignancy, to initiate further diagnostics.

Several investigators studied ferritin in unspecified, non-ICU or non-exclusively ICU patients [13,14,23,24,33,45]. Schram et al. [14], Fauter et al. [24], Senjo et al. [13], and Sackett et al. [23] contained both ICU and non-ICU patients, but without any separate analysis for ICU patients. Schram et al. [14] were the only investigators who performed multivariable analysis. Our results differ from their study, which could be explained by a significantly lower threshold for ferritin as an inclusion criterion, a complete focus on ICU patients, a comprehensive separation of all ICD-10 codes into 17 different disease groups, and the currently largest available cohort of mixed ICU patients with measured ferritin in our study.

Our study has several limitations. Since this is a retrospective study, only available data could be analyzed. Furthermore, the study contains only patients admitted to one of the ICUs and also those who had a ferritin measurement during their ICU stay, leaving a risk of selection bias. Some patients could be more likely to receive ferritin measurements than others, so that several disease groups might be over- or underrepresented. In this regard, we cannot exclude the possibility that some of the non-HLH patients indeed had a hematologically triggered HLH or a liver dysfunction due to HLH.

## 5. Conclusions

In this first study systematically investigating differential diagnoses of hyperferritinemia in adult critically ill patients without HLH, we found sepsis or septic shock, liver disease (except hepatitis), and hematological malignancy to be associated with hyperferritinemia. Together with HLH, they complete the quartet of important differential diagnoses of hyperferritinemia in adult critically ill patients. When present in combination, ferritin levels were further increased. As these conditions are also related to HLH, it is important to apply HLH-2004 criteria for exclusion of HLH in hyperferritinemic patients. Hyperferritinemic critically ill patients without HLH require further screening for the differential diagnoses, such as sepsis or septic shock, liver disease, and hematological malignancy. High mortality rates in critically ill patients with hyperferritinemia underline the importance of quickly finding the underlying condition. Our study provides a contribution to the correct interpretation of hyperferritinemia in critical care.

## Figures and Tables

**Figure 1 jcm-12-00192-f001:**
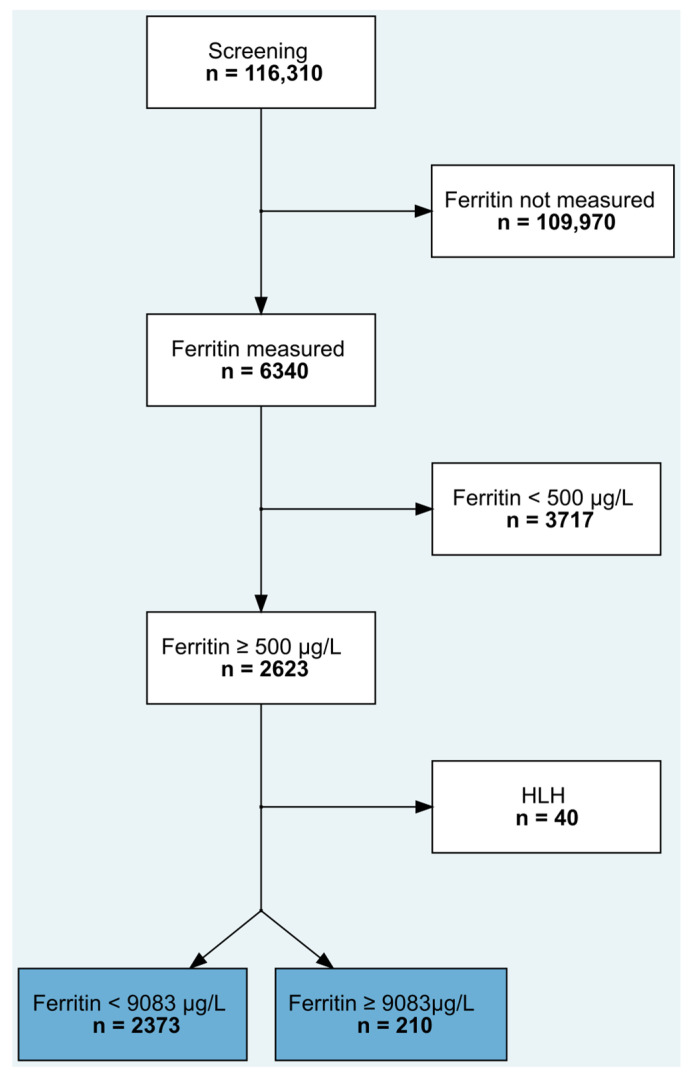
Consort flow diagram. HLH, hemophagocytic lymphohistiocytosis.

**Figure 2 jcm-12-00192-f002:**
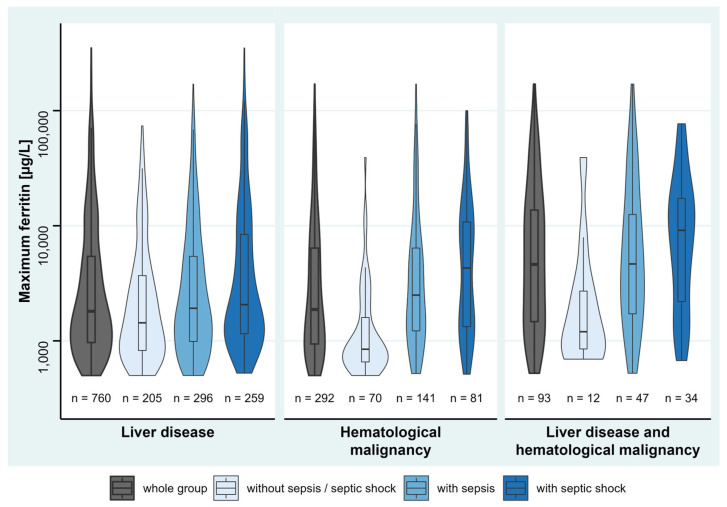
Ferritin levels of positively associated disease groups (each and in combination) on maximum ferritin, stratified by sepsis status. Violin plots show data distribution and enclosed boxplots show median with quartiles (25–75%). Whiskers below and above show minimum and maximum values, respectively. For improved visualization, *y*-axis is plotted in logarithmic scale. Due to various numbers of ICD-10 codes in each single patient, patients partially overlap between the disease groups.

**Figure 3 jcm-12-00192-f003:**
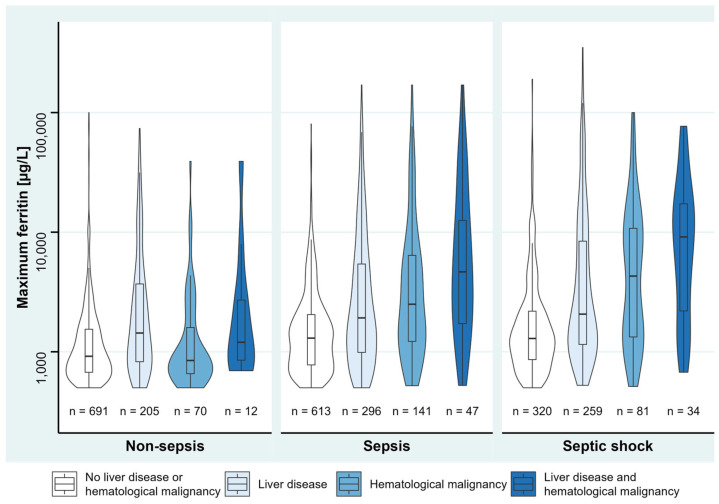
Ferritin levels of sepsis groups stratified by the presence of liver disease and hematological malignancy. Violin plots show data distribution and enclosed boxplots show median with quartiles (25–75%). Whiskers below and above show minimum and maximum values, respectively. For improved visualization, *y*-axis is plotted in logarithmic scale. Due to various numbers of ICD-10 codes in each single patient, patients partially overlap between the disease groups.

**Figure 4 jcm-12-00192-f004:**
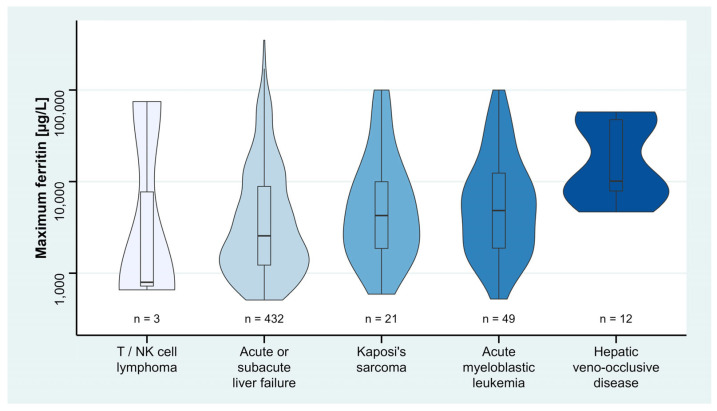
Ferritin levels of positively associated subgroups of liver disease and hematological malignancy on maximum ferritin. Violin plots show data distribution and enclosed boxplots show median with quartiles (25–75%). Whiskers below and above show minimum and maximum values, respectively. For improved visualization, *y*-axis is plotted in logarithmic scale. Due to various numbers of ICD-10 codes in each single patient, patients partially overlap between the groups. NK, natural killer.

**Figure 5 jcm-12-00192-f005:**
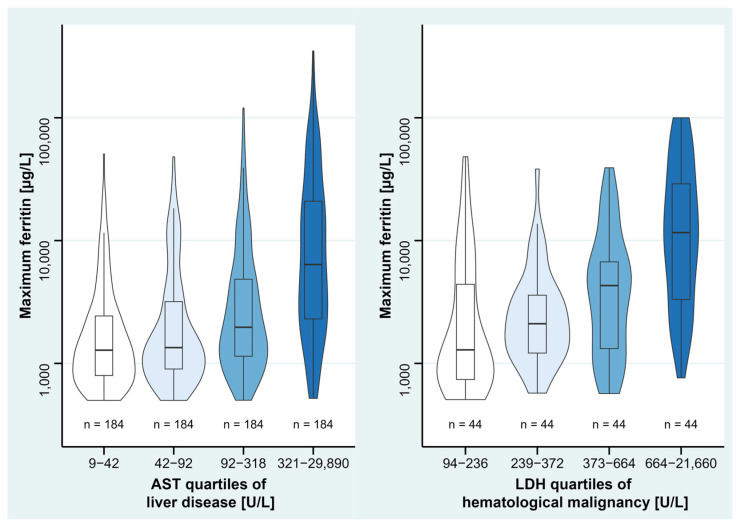
Ferritin levels of liver disease and hematological malignancy, divided into quartiles of their diagnostic markers. Violin plots show data distribution and enclosed boxplots show median with quartiles (25–75%). Whiskers below and above show minimum and maximum values, respectively. For improved visualization, *y*-axis is plotted in logarithmic scale. Due to various numbers of ICD-10 codes in each single patient, patients partially overlap between the disease groups. AST, aspartate aminotransferase. LDH, lactate dehydrogenase.

**Table 1 jcm-12-00192-t001:** Patient characteristics, diagnostic markers and outcome parameters.

	Total (n = 2583)	Ferritin <9083 µg/L (n = 2373)	Ferritin ≥9083 µg/L (n = 210)	*p* Value
Age [years]	62 (49–73)	63 (50–73)	52 (37–64)	<0.001 ‡
Male sex [n] (%)	1588 (61.5%)	1474 (62.1%)	114 (54.3%)	0.025 †
Body Mass Index [kg/m^2^]	25.0 (22.0–29.0)	25.0 (22.0–29.0)	24.1 (21.0–29.0)	0.121 ‡
Maximum ferritin [µg/L]	1289 (784–2480)	1190 (757–1997)	16,964 (12,127–37,406)	<0.001 ‡
Sepsis without shock [n] (%)	1003 (38.8%)	928 (39.1%)	75 (35.7%)	0.334 †
Septic shock [n] (%)	626 (24.2%)	538 (22.7%)	88 (41.9%)	<0.001 †
Tuberculosis [n] (%)	20 (0.8%)	18 (0.8%)	2 (1.0%)	0.759 †
Hepatitis [n] (%)	159 (6.2%)	139 (5.9%)	20 (9.5%)	0.034 †
VZV [n] (%)	31 (1.2%)	29 (1.2%)	2 (1.0%)	0.731 †
Malaria [n] (%)	4 (0.2%)	3 (0.1%)	1 (0.5%)	0.217 †
HSV [n] (%)	107 (4.1%)	98 (4.1%)	9 (4.3%)	0.913 †
Influenza [n] (%)	58 (2.2%	52 (2.2%)	6 (2.9%)	0.532 †
Liver disease (except hepatitis) [n] (%)	760 (29.4%)	619 (26.1%)	141 (67.1%)	<0.001 †
History of stem cell/organ transplantation [n] (%)	316 (12.2%)	272 (11.5%)	44 (21.0%)	<0.001 †
CMV [n] (%)	139 (5.4%)	118 (5.0%)	21 (10.0%)	0.002 †
HIV [n] (%)	70 (2.7%)	62 (2.6%)	8 (3.8%)	0.306
EBV [n] (%)	40 (1.5%)	34 (1.4%)	6 (2.9%)	0.109 †
Renal disease [n] (%)	1820 (70.5%)	1674 (70.5%)	146 (69.5%)	0.756 †
(Bacterial/viral/fungal) infection [n] (%)	2250 (87.1%)	2067 (87.1%)	183 (87.1%)	0.987 †
Inflammation without infection [n] (%)	843 (32.6%)	790 (33.3%)	53 (25.2%)	0.017 †
Autoimmune disease [n] (%)	185 (7.2%)	172 (7.2%)	13 (6.2%)	0.569 †
Solid malignancy [n] (%)	428 (16.6%)	401 (16.9%)	27 (12.9%)	0.131 †
Hematological malignancy [n] (%)	292 (11.3%)	239 (10.1%)	53 (25.2%)	<0.001 †
Pre-existing immunosuppression [n] (%)	728 (28.2%)	638 (26.9%)	90 (42.9%)	<0.001 ‡
Hemoglobin [g/dL]	8.8 (7.9–9.9)	8.8 (7.9–9.9)	8.9 (7.8–10.1)	0.807 ‡
Platelet count [/nL]	170 (88–270)	179 (101–278)	56 (25–136)	<0.001 ‡
Leukocyte count [/nL]	9.0 (6.1–13.3)	9.1 (6.2–13.2)	8.5 (3.6–15.9)	0.057 ‡
Triglycerides [mg/dL], n = 765|90	158 (104–247)	155 (103–233)	218 (121–420)	<0.001 ‡
Fibrinogen [mg/dL], n = 991|153	3.8 (2.5–5.3)	4.0 (2.7–5.3)	2.9 (1.8–4.5)	<0.001 ‡
INR, n = 2341|209	1.3 (1.2–1.5)	1.3 (1.2–1.4)	1.6 (1.3–2.3)	<0.001 ‡
aPTT [sec], n = 2347|209	45.7 (38.8–55.9)	45.1 (38.5–55.1)	51.1 (43.7–64.0)	<0.001 ‡
AST [U/L], n = 2120|207	47 (26–108)	43 (25–89)	350 (91–2955)	<0.001 ‡
ALT [U/L], n = 2107|208	34 (18–79)	32 (17–66)	207 (47–1478)	<0.001 ‡
Bilirubin [mg/dl], n = 2050|198	0.8 (0.4–2.4)	0.7 (0.4–2.0)	3.3 (1.3–9.2)	<0.001 ‡
ɣGT [U/l], n = 2082|208	124 (55–272)	121 (54–266)	157 (76–375)	<0.001 ‡
AP [U/l], n = 1758|205	138 (86–245)	134 (84–230)	212 (114–455)	<0.001 ‡
Albumin [g/L], n = 1779|195	26 (22.3–30.2)	26.0 (22.4–30.3)	25.4 (21.6–29.8)	0.060 ‡
Creatinine [mg/dl], n = 2254|170	1.3 (0.7–2.6)	1.3 (0.7–2.7)	1.7 (0.8–2.6)	0.112 ‡
CRP [mg/L], n = 2306|208	81.5 (38.4–155.8)	79 (37–150)	119 (56–222)	<0.001 ‡
PCT [µg/L], n = 1446|165	1.0 (0.4–3.6)	0.8 (0.3–2.7)	3.6 (1.5–12.4)	<0.001 ‡
Lactate [mg/dl], n = 2347|207	13 (9–20)	13 (9–19)	25 (12–60)	<0.001 ‡
LDH [U/L], n = 1142|155	361 (246–593)	339 (237–450)	1175 (489–2880)	<0.001 ‡
Max. core body temperature [°C], n = 2250|202	38.2 (37.5–38.9)	38.2 (37.5–38.9)	38.3 (37.6–39.1)	0.033 ‡
Hemodialysis [n] (%)	1357 (52.5%)	1219 (51.4%)	138 (65.7%)	<0.001 †
ECLA/ECMO [n] (%)	188 (7.3%)	168 (7.1%)	20 (9.5%)	0.191 †
ICU admission SOFA score	6 (3–9)	6 (3–9)	8 (4–12)	<0.001 ‡
Maximum SOFA score	11 (7–15)	11 (7–15)	15 (9–19)	<0.001 ‡
ICU duration [d]	19 (6–47)	20 (6–47)	15 (5–43)	0.047 ‡
Inpatient duration [d]	38 (18–76)	39 (19–77)	31 (12–74)	0.009 ‡
Mortality [n] (%)	741 (28.7%)	626 (26.4%)	115 (54.8%)	<0.001 †

Continuous quantities in median with quartiles, categorical parameters with count and percentage. Parameters with n representing the number of patients with available data in each group, if not available in all patients. ALT, alanine aminotransferase. AP, alkaline phosphatase. aPTT, activated partial thromboplastin time. AST, aspartate aminotransferase. CMV, cytomegalovirus. CRP, c reactive protein. EBV, Epstein-Barr virus. ECLA, extracorporeal lung assist. ECMO, extracorporeal membrane oxygenation. ɣGT, gamma glutamyl transferase. HIV, human immunodeficiency virus. HSV, herpes simplex virus. ICU, intensive care unit. INR, international normalized ratio. LDH, lactate dehydrogenase. PCT, procalcitonin. SOFA, sequential organ failure assessment. VZV, varicella-zoster virus. *p* values calculated using the Mann-Whitney-U test ^‡^ or the χ2 test ^†^ as appropriate. Due to various numbers of ICD-10 codes in each single patient, patients partially overlap between the disease groups.

**Table 2 jcm-12-00192-t002:** Multivariable linear regression analysis for the influence of underlying diseases on maximum ferritin levels.

Covariates	Regression Coefficient	95% CI	*p* Value
Age	−38.6	−70.7, −6.4	0.019
Sex (male)	62.0	−994.6, 1118.5	0.908
Body Mass Index	−28.1	−104.0, 47.8	0.468
Maximum SOFA score	190.9	82.0, 299.8	<0.001
Sepsis or septic shock *	903.6	129.9, 1677.3	0.022
Pre-existing immunosuppression	965.8	−352.8, 2284.4	0.151
Liver disease	4068.3	2861.8, 5274.8	<0.001
History of stem cell/organ transplantation	134.6	−1656.7, 1925.9	0.883
Hematological malignancy	3148.0	1504.0, 4792.0	<0.001

Multivariable linear regression analysis was performed with maximum ferritin as the dependent variable in all 2583 patients (adjusted Nagelkerke R^2^ = 0.053). * Trichotomous categorical variable, classified as non-sepsis (reference), sepsis, and septic shock. CI, confidence interval. SOFA, sequential organ failure assessment.

**Table 3 jcm-12-00192-t003:** Multivariable linear regression analysis for the influence of subgroups of liver disease and hematological malignancy on maximum ferritin.

Covariates	Regression Coefficient	95% CI	*p* Value
Age	−30.6	−62.3, 1.1	0.059
Sex (male)	−157.9	−1200.9, 885.0	0.767
Body Mass Index	−22.2	−97.0, 52.6	0.560
Maximum SOFA score	135.4	26.7, 244.0	0.015
Sepsis or septic shock *	935.2	168.0, 1702.4	0.017
Pre-existing immunosuppression	893.3	−398.9, 2185.6	0.175
Registration for high-urgency liver transplantation (Z75.77)	−71.6	−5719.9, 5576.6	0.980
Acute or subacute liver failure (K72.0)	7539.9	5538.9, 9541.0	<0.001
Chronic liver failure (K72.1)	4695.8	−1022.0, 10,413.6	0.107
Grade one hepatic encephalopathy (K72.71)	2890.7	−1343.2, 7124.6	0.181
Grade two hepatic encephalopathy (K72.72)	−1742.8	−5975.2, 2489.6	0.419
Grade three hepatic encephalopathy (K72.73)	−1078.2	−4925.2, 2768.9	0.583
Grade four hepatic encephalopathy (K72.74)	951.7	−2413.3, 4316.7	0.579
Hepatic encephalopathy, grade unspecified (K72.79)	−3277.5	−5966.3, −588.7	0.017
Hepatic veno-occlusive disease (K76.5)	15,783.9	8237.4, 23,330.4	<0.001
History of stem cell/ organ transplantation	70.3	−1732.5, 1873.1	0.939
T/NK cell lymphoma (C84.5)	22,919.1	8103.3, 37,734.9	0.002
Acute lymphocytic leukemia (C91.00)	5093.2	−2408.3, 12,594.7	0.183
Acute myeloblastic leukemia (C92.00)	5235.3	1475.0, 8995.6	0.006
Kaposi’s sarcoma (D46.7)	9118.6	3485.2, 14,752.0	0.002
Aplastic anemia (D61.9)	4083.2	−185.2, 8351.7	0.061

Multivariable linear regression analysis was performed with maximum ferritin as the dependent variable in all 2583 patients (adjusted Nagelkerke R^2^ = 0.082). * Trichotomous categorical variable classified as non-sepsis (reference), sepsis, and septic shock. CI, confidence interval. NK, natural killer. SOFA, sequential organ failure assessment.

## Data Availability

Due to legal restrictions imposed by the data protection commissioner of the Charité–Universitätsmedizin Berlin, public sharing of study data with other researchers or entities is restricted to anonymized data. Requests may be sent to dai-researchdata@charite.de.

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
