# Peer review of "Differential Diagnosis of Hyperferritinemia in Critically Ill Patients"

_jcm, 2022, doi:10.3390/jcm12010192_

Round 1
Reviewer 1 Report
Thanks you very much for the opportunity to review this manuscript, which is trying to understand if hyperferritinaemia, which does not meet HLH definition is associated with other diseases important for intensive care providers.
I have several questions and comments to the authors.
The first major comment is that it's unclear to the reader, what exactly was the purpose of the study. From the introduction it seems it was to see which ICD-10 diseases codes are associated with high ferritin levels. Despite the introduction I still struggle to understand how this knowledge can help the bedside clinicians?
The second major comment is regarding the methods of the study. The observation period is quite long and I wonder if there have been a change in the frequency of ferritin measurements over time. I would be interested to know, how the distribution of patient numbers was over the recruitment period? Is it possible, that the authors have an implicit selection bias, that they only included patients where the clinicians thought the haemopoesis is impaired, or there is hyperinflammation present? It would be useful therefore to understand, how many of the patients diagnosed with the certain ICD-codes had ferritin levels measured.
The authors should provide the used ICD-10 codes as an appendix to the current paper, referencing a previous supplement is not enough in my opinion.
I have significant problems with regression models, more so with the post-hoc one, however the next comment applies to both. The authors would need to assure the readers that their model is generalisable and not overfitting the data. Goodness-of-fit parameters should be provided. In the post-hoc model, the authors included far too many parameters for a smallish sample size. They should provide an explanation how this model was built and how they have followed best practice methods to achieve parsimony for the parameters, which is necessary given the low event rate and small sample size.
When looking at the regression analysis data, one can't escape the thought that several laboratory parameters showing effect on ferritin levels could be subsequently used in the diagnosis of certain diseases or syndromes. How did the authors deal with this issue?
I found the box and whisker plots on the logarithmic scale very difficult to interpret. I was wondering if the use of violin plots could help to describe the data and it's distribution a lot better?
Because of the lack of clearly communicated research question, I found the discussion to be difficult to follow. I wonder if the authors could simplify the paper by removing the somewhat spurious post-hoc analysis and the resulting discussion elements and concentrate on their main question.
Reviewer 2 Report
This work aimed at analyzing the predictive factors for hyperferritinemia in critically Ill Patients without Hemophagocytic Lymphohistiocytosis. However, the analyzing procedure was not well designed. Therefore, the conclusion was not solid enough.
In the first part, 2623 patients with hyperferritinemia were grouped into hyperferritinemia and extreme hyperferritinemia groups according to the ferritin levels. The clinical and laboratory examinations were compared between the two groups.
In the following part, multivariable regression analysis was used to correlate the variables to the maximum ferritin level. How did the authors select variables for multivariable regression analysis? Did the authors consider using univariable regression firstly to identify variables associated with maximum ferritin level and utilize the identified variables to perform the multivariable regression analysis?
For diagnostic biomarkers, it is suggested that the patients be separated into hyperferritinemia and non-hyperferritinemia groups.
Overall, it is suggested that the authors reconsider the methods for analyzing the data. Using more precise methods may achieve a more profound insight into the understanding of the intended purpose of this study.
Round 2
Reviewer 2 Report
The revised manuscript has addressed all of my comments.